# Implicit Posterior Variational Inference for Deep Gaussian Processes

**Haibin Yu**[*], **Yizhou Chen**[*]**, Zhongxiang Dai, Bryan Kian Hsiang Low, and Patrick Jaillet**[†]

Dept. of Computer Science, National University of Singapore, Republic of Singapore

Dept. of Electrical Engineering and Computer Science, MIT, USA[†]

{haibin,ychen041,daiz,lowkh}@comp.nus.edu.sg, jaillet@mit.edu[†]

## Abstract

A multi-layer *deep Gaussian process* (DGP) model is a hierarchical composition of GP models with a greater expressive power. Exact DGP inference is intractable, which has motivated the recent development of deterministic and stochastic approximation methods. Unfortunately, the deterministic approximation methods yield a biased posterior belief while the stochastic one is computationally costly. This paper presents an *implicit posterior variational inference* (IPVI) framework for DGPs that can ideally recover an unbiased posterior belief and still preserve time efficiency. Inspired by generative adversarial networks, our IPVI framework achieves this by casting the DGP inference problem as a two-player game in which a Nash equilibrium, interestingly, coincides with an unbiased posterior belief. This consequently inspires us to devise a best-response dynamics algorithm to search for a Nash equilibrium (i.e., an unbiased posterior belief). Empirical evaluation shows that IPVI outperforms the state-of-the-art approximation methods for DGPs.

## 1   Introduction

The expressive power of the Bayesian non-parametric *Gaussian process* (GP) [46] models can be significantly boosted by composing them hierarchically into a multi-layer *deep GP* (DGP) model, as shown in the seminal work of [12]. Though the DGP model can likewise exploit the notion of inducing variables [5, 24, 25, 36, 40, 45, 55, 57] to improve its scalability, doing so does not immediately entail tractable inference, unlike the GP model. This has motivated the development of deterministic and stochastic approximation methods, the former of which have imposed varying structural assumptions across the DGP hidden layers and assumed a Gaussian posterior belief of the inducing variables [3, 10, 12, 20, 48]. However, the work of [18] has demonstrated that with at least one DGP hidden layer, the posterior belief of the inducing variables is usually non-Gaussian, hence potentially compromising the performance of the deterministic approximation methods due to their biased posterior belief. To resolve this, the stochastic approximation method of [18] utilizes *stochastic gradient Hamiltonian Monte Carlo* (SGHMC) sampling to draw unbiased samples from the posterior belief. But, generating such samples is computationally costly in both training and prediction due to its sequential sampling procedure [54] and its convergence is also difficult to assess. So, the challenge remains in devising a time-efficient approximation method that can recover an unbiased posterior belief.

This paper presents an *implicit posterior variational inference* (IPVI) framework for DGPs (Section 3) that can ideally recover an unbiased posterior belief and still preserve time efficiency, hence combining the best of both worlds (respectively, stochastic and deterministic approximation methods). Inspired by generative adversarial networks [17] that can generate samples to represent complex distributions

---

[*]Equal contribution.

which are hard to model using an explicit likelihood [31, 53], our IPVI framework achieves this by casting the DGP inference problem as a two-player game in which a Nash equilibrium, interestingly, coincides with an unbiased posterior belief. This consequently inspires us to devise a best-response dynamics algorithm to search for a Nash equilibrium [2] (i.e., an unbiased posterior belief). In Section 4, we discuss how the architecture of the generator in our IPVI framework is designed to enable parameter tying for a DGP model to alleviate overfitting. We empirically evaluate the performance of IPVI on several real-world datasets in supervised (e.g., regression and classification) and unsupervised learning tasks (Section 5).

## 2 Background and Related Work

**Gaussian Process (GP).** Let a random function $f : \mathbb{R}^D \to \mathbb{R}$ be distributed by a GP with a zero prior mean and covariance function $k : \mathbb{R}^D \times \mathbb{R}^D \to \mathbb{R}$. That is, suppose that a set $\mathbf{y} \triangleq \{y_n\}_{n=1}^N$ of $N$ noisy observed outputs $y_n \triangleq f(\mathbf{x}_n) + \varepsilon(\mathbf{x}_n)$ (i.e., corrupted by an i.i.d. Gaussian noise $\varepsilon(\mathbf{x}_n)$ with noise variance $\nu^2$) are available for some set $\mathbf{X} \triangleq \{\mathbf{x}_n\}_{n=1}^N$ of $N$ training inputs. Then, the set $\mathbf{f} \triangleq \{f(\mathbf{x}_n)\}_{n=1}^N$ of latent outputs follow a Gaussian prior belief $p(\mathbf{f}) \triangleq \mathcal{N}(\mathbf{f}|\mathbf{0}, \mathbf{K_{XX}})$ where $\mathbf{K_{XX}}$ denotes a covariance matrix with components $k(\mathbf{x}_n, \mathbf{x}_{n'})$ for $n, n' = 1, \ldots, N$. It follows that $p(\mathbf{y}|\mathbf{f}) = \mathcal{N}(\mathbf{y}|\mathbf{f}, \nu^2\mathbf{I})$. The GP predictive/posterior belief of the latent outputs $\mathbf{f}^\star \triangleq \{f(\mathbf{x}^\star)\}_{\mathbf{x}^\star \in \mathbf{X}^\star}$ for any set $\mathbf{X}^\star$ of test inputs can be computed in closed form [46] by marginalizing out $\mathbf{f}$: $p(\mathbf{f}^\star|\mathbf{y}) = \int p(\mathbf{f}^\star|\mathbf{f}) \, p(\mathbf{f}|\mathbf{y}) \, \mathrm{d}\mathbf{f}$ but incurs cubic time in $N$, hence scaling poorly to massive datasets.

To improve its scalability to linear time in $N$, the *sparse GP* (SGP) models spanned by the unifying view of [45] exploit a set $\mathbf{u} \triangleq \{u_m \triangleq f(\mathbf{z}_m)\}_{m=1}^M$ of inducing output variables for some small set $\mathbf{Z} \triangleq \{\mathbf{z}_m\}_{m=1}^M$ of inducing inputs (i.e., $M \ll N$). Then,

$$p(\mathbf{y}, \mathbf{f}, \mathbf{u}) = p(\mathbf{y}|\mathbf{f}) \, p(\mathbf{f}|\mathbf{u}) \, p(\mathbf{u}) \tag{1}$$

such that $p(\mathbf{f}|\mathbf{u}) = \mathcal{N}(\mathbf{f}|\mathbf{K_{XZ}}\mathbf{K_{ZZ}}^{-1}\mathbf{u}, \mathbf{K_{XX}} - \mathbf{K_{XZ}}\mathbf{K_{ZZ}}^{-1}\mathbf{K_{ZX}})$ where, with a slight abuse of notation, $\mathbf{u}$ is treated as a column vector here, $\mathbf{K_{XZ}} \triangleq \mathbf{K_{ZX}}^\top$, and $\mathbf{K_{ZZ}}$ and $\mathbf{K_{ZX}}$ denote covariance matrices with components $k(\mathbf{z}_m, \mathbf{z}_{m'})$ for $m, m' = 1, \ldots, M$ and $k(\mathbf{z}_m, \mathbf{x}_n)$ for $m = 1, \ldots, M$ and $n = 1, \ldots, N$, respectively. The SGP predictive belief can also be computed in closed form by marginalizing out $\mathbf{u}$: $p(\mathbf{f}^\star|\mathbf{y}) = \int p(\mathbf{f}^\star|\mathbf{u}) \, p(\mathbf{u}|\mathbf{y}) \, \mathrm{d}\mathbf{u}$.

The work of [50] has proposed a principled *variational inference* (VI) framework that approximates the joint posterior belief $p(\mathbf{f}, \mathbf{u}|\mathbf{y})$ with a variational posterior $q(\mathbf{f}, \mathbf{u}) \triangleq p(\mathbf{f}|\mathbf{u}) \, q(\mathbf{u})$ by minimizing the *Kullback-Leibler* (KL) distance between them, which is equivalent to maximizing a lower bound of the log-marginal likelihood (i.e., also known as the *evidence lower bound* (ELBO)):

$$\mathrm{ELBO} \triangleq \mathbb{E}_{q(\mathbf{f})}[\log p(\mathbf{y}|\mathbf{f})] - \mathrm{KL}[q(\mathbf{u})\|p(\mathbf{u})]$$

where $q(\mathbf{f}) \triangleq \int p(\mathbf{f}|\mathbf{u}) \, q(\mathbf{u}) \, \mathrm{d}\mathbf{u}$. A common choice in VI is the Gaussian variational posterior $q(\mathbf{u}) \triangleq \mathcal{N}(\mathbf{u}|\mathbf{m}, \mathbf{S})$ of the inducing variables $\mathbf{u}$ [14, 16, 19, 24, 25, 51] which results in a Gaussian marginal $q(\mathbf{f}) = \mathcal{N}(\mathbf{f}|\boldsymbol{\mu}, \boldsymbol{\Sigma})$ where $\boldsymbol{\mu} \triangleq \mathbf{K_{XZ}}\mathbf{K_{ZZ}}^{-1}\mathbf{m}$ and $\boldsymbol{\Sigma} \triangleq \mathbf{K_{XX}} - \mathbf{K_{XZ}}\mathbf{K_{ZZ}}^{-1}(\mathbf{K_{ZZ}} - \mathbf{S})\mathbf{K_{ZZ}}^{-1}\mathbf{K_{ZX}}$.

**Deep Gaussian Process (DGP).** A multi-layer DGP model is a hierarchical composition of GP models. Consider a DGP with a depth of $L$ such that each DGP layer is associated with a set $\mathbf{F}_{\ell-1}$ of inputs and a set $\mathbf{F}_\ell$ of outputs for $\ell = 1, \ldots, L$ and $\mathbf{F}_0 \triangleq \mathbf{X}$. Let $\mathcal{F} \triangleq \{\mathbf{F}_\ell\}_{\ell=1}^L$, and the inducing inputs and corresponding inducing output variables for DGP layers $\ell = 1, \ldots, L$ be denoted by the respective sets $\mathcal{Z} \triangleq \{\mathbf{Z}_\ell\}_{\ell=1}^L$ and $\mathcal{U} \triangleq \{\mathbf{U}_\ell\}_{\ell=1}^L$. Similar to the joint probability distribution of the SGP model (1),

$$p(\mathbf{y}, \mathcal{F}, \mathcal{U}) = \underbrace{p(\mathbf{y}|\mathbf{F}_L)}_{\text{data likelihood}} \underbrace{\left[\prod_{\ell=1}^L p(\mathbf{F}_\ell|\mathbf{U}_\ell)\right] p(\mathcal{U})}_{\text{DGP prior}}.$$

Similarly, the variational posterior is assumed to be $q(\mathcal{F}, \mathcal{U}) \triangleq \left[\prod_{\ell=1}^L p(\mathbf{F}_\ell|\mathbf{U}_\ell)\right] q(\mathcal{U})$, thus resulting in the following ELBO for the DGP model:

$$\mathrm{ELBO} \triangleq \int q(\mathbf{F}_L) \log p(\mathbf{y}|\mathbf{F}_L) \, \mathrm{d}\mathbf{F}_L - \mathrm{KL}[q(\mathcal{U})\|p(\mathcal{U})] \tag{2}$$

where $q(\mathbf{F}_L) \triangleq \int \prod_{\ell=1}^{L} p(\mathbf{F}_\ell | \mathbf{U}_\ell, \mathbf{F}_{\ell-1}) \, q(\mathcal{U}) \, \mathrm{d}\mathbf{F}_1 \ldots \mathrm{d}\mathbf{F}_{L-1} \mathrm{d}\mathcal{U}$. To compute $q(\mathbf{F}_L)$, the work of [48] has proposed the use of the reparameterization trick [32] and Monte Carlo sampling, which are adopted in this work.

*Remark* 1. To the best of our knowledge, the DGP models exploiting the inducing variables[2] and the VI framework [10, 12, 20, 48] have imposed the highly restrictive assumptions of (i) mean field approximation $q(\mathcal{U}) \triangleq \prod_{\ell=1}^{L} q(\mathbf{U}_\ell)$ and (ii) biased Gaussian variational posterior $q(\mathbf{U}_\ell)$. In fact, the true posterior belief usually exhibits a high correlation across the DGP layers and is non-Gaussian [18], hence potentially compromising the performance of such deterministic approximation methods for DGP models. To remove these assumptions, we will propose a principled approximation method that can generate unbiased posterior samples even under the VI framework, as detailed in Section 3.

## 3   Implicit Posterior Variational Inference (IPVI) for DGPs

Unlike the conventional VI framework for existing DGP models [10, 12, 20, 48], our proposed IPVI framework does not need to impose their highly restrictive assumptions (Remark 1) and can still preserve the time efficiency of VI. Inspired by previous works on adversarial-based inference [30, 42], IPVI achieves this by first generating posterior samples $\mathcal{U} \triangleq g_\Phi(\epsilon)$ with a black-box **generator** $g_\Phi(\epsilon)$ parameterized by $\Phi$ and a random noise $\epsilon \sim \mathcal{N}(\mathbf{0}, \mathbf{I})$. By representing the variational posterior as $q_\Phi(\mathcal{U}) \triangleq \int p(\mathcal{U}|\epsilon)\mathrm{d}\epsilon$, the ELBO in (2) can be re-written as

$$\text{ELBO} = \mathbb{E}_{q(\mathbf{F}_L)}[\log p(\mathbf{y}|\mathbf{F}_L)] - \text{KL}[q_\Phi(\mathcal{U})\|p(\mathcal{U})] . \tag{3}$$

An immediate advantage of the generator $g_\Phi(\epsilon)$ is that it can generate the posterior samples in parallel by feeding it a batch of randomly sampled $\epsilon$'s. However, representing the variational posterior $q_\Phi(\mathcal{U})$ implicitly makes it impossible to evaluate the KL distance in (3) since $q_\Phi(\mathcal{U})$ cannot be calculated explicitly. By observing that the KL distance is equal to the expectation of the log-density ratio $\mathbb{E}_{q_\Phi(\mathcal{U})}[\log q_\Phi(\mathcal{U}) - \log p(\mathcal{U})]$, we can circumvent an explicit calculation of the KL distance term by implicitly representing the log-density ratio as a separate function $T$ to be optimized, as shown in our first result below:

**Proposition 1.** *Let* $\sigma(x) \triangleq 1/(1 + \exp(-x))$. *Consider the following maximization problem:*

$$\max_T \ \mathbb{E}_{p(\mathcal{U})}[\log(1 - \sigma(T(\mathcal{U})))] + \mathbb{E}_{q_\Phi(\mathcal{U})}[\log \sigma(T(\mathcal{U}))] . \tag{4}$$

*If* $p(\mathcal{U})$ *and* $q_\Phi(\mathcal{U})$ *are known, then the optimal* $T^*$ *with respect to* (4) *is the log-density ratio:*

$$T^*(\mathcal{U}) = \log q_\Phi(\mathcal{U}) - \log p(\mathcal{U}) . \tag{5}$$

Its proof (Appendix A) is similar to that of Proposition 1 in [17] except that we use a sigmoid function $\sigma$ to reveal the log-density ratio. Note that (4) defines a binary cross-entropy between samples from the variational posterior $q_\Phi(\mathcal{U})$ and prior $p(\mathcal{U})$. Intuitively, $T$ in (4), which we refer to as a **discriminator**, tries to distinguish between $q_\Phi(\mathcal{U})$ and $p(\mathcal{U})$ by outputting $\sigma(T(\mathcal{U}))$ as the probability of $\mathcal{U}$ being a sample from $q_\Phi(\mathcal{U})$ rather than $p(\mathcal{U})$.

Using Proposition 1 (i.e., (5)), the ELBO in (3) can be re-written as

$$\text{ELBO} = \mathbb{E}_{q_\Phi(\mathcal{U})}[\mathcal{L}(\theta, \mathbf{X}, \mathbf{y}, \mathcal{U}) - T^*(\mathcal{U})] \tag{6}$$

where $\mathcal{L}(\theta, \mathbf{X}, \mathbf{y}, \mathcal{U}) \triangleq \mathbb{E}_{p(\mathbf{F}_L|\mathcal{U})}[\log p(\mathbf{y}|\mathbf{F}_L)]$ and $\theta$ denotes the DGP model hyperparameters. The ELBO can now be calculated given the optimal discriminator $T^*$. In our implementation, we adopt a parametric representation for discriminator $T$. In principle, the parametric representation is required to be expressive enough to be able to represent the optimal discriminator $T^*$ accurately. Motivated by the fact that deep neural networks are universal function approximators [29], we represent discriminator $T_\Psi$ by a neural network with parameters $\Psi$; the optimal $T_{\Psi^*}$ is thus parameterized by $\Psi^*$. The architecture of the generator and discriminator in our IPVI framework will be discussed in Section 4.

The ELBO in (6) can be optimized with respect to $\Phi$ and $\theta$ via gradient ascent, provided that the optimal $T_{\Psi^*}$ (with respect to $q_\Phi$) can be obtained in every iteration. One way to achieve this is to cast

**Algorithm 1: Main**

1 Randomly initialize $\theta$, $\Psi$, $\Phi$
2 **while** *not converged* **do**
3 $\quad$ Run Algorithm 2
4 $\quad$ Run Algorithm 3

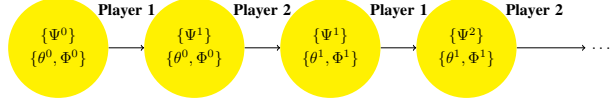

**Algorithm 2: Player 1**

1 Sample $\{\boldsymbol{\mathcal{V}}_1, \ldots, \boldsymbol{\mathcal{V}}_K\}$ from $p(\boldsymbol{\mathcal{U}})$
2 Sample $\{\boldsymbol{\mathcal{U}}_1, \ldots, \boldsymbol{\mathcal{U}}_K\}$ from $q_\Phi(\boldsymbol{\mathcal{U}})$
3 Compute gradient w.r.t. $\Psi$ from (7):

4
$$g_\Psi \triangleq \nabla_\Psi\left[\frac{1}{K}\sum_{k=1}^{K}\log(1-\sigma(T_\Psi(\boldsymbol{\mathcal{V}}_k)))\right]$$
$$+\nabla_\Psi\left[\frac{1}{K}\sum_{k=1}^{K}\log\sigma(T_\Psi(\boldsymbol{\mathcal{U}}_k))\right]$$

5 SGA update for $\Psi$:
6 $\quad \Psi \leftarrow \Psi + \alpha_\Psi\, g_\Psi$

**Algorithm 3: Player 2**

1 Sample mini-batch $(\mathbf{X}_b, \mathbf{y}_b)$ from $(\mathbf{X}, \mathbf{y})$
2 Sample $\{\boldsymbol{\mathcal{U}}_1, \ldots, \boldsymbol{\mathcal{U}}_K\}$ from $q_\Phi(\boldsymbol{\mathcal{U}})$
3 Compute gradients w.r.t. $\theta$ and $\Phi$ from (7):

4
$$g_\theta \triangleq \nabla_\theta\left[\frac{1}{K}\sum_{k=1}^{K}\mathcal{L}(\theta, \mathbf{X}_b, \mathbf{y}_b, \boldsymbol{\mathcal{U}}_k)\right]$$
$$g_\Phi \triangleq \nabla_\Phi\left[\frac{1}{K}\sum_{k=1}^{K}\mathcal{L}(\theta, \mathbf{X}_b, \mathbf{y}_b, \boldsymbol{\mathcal{U}}_k)-T_\Psi(\boldsymbol{\mathcal{U}}_k)\right]$$

5 SGA updates for $\theta$ and $\Phi$:
6 $\quad \theta \leftarrow \theta + \alpha_\theta\, g_\theta\,, \quad \Phi \leftarrow \Phi + \alpha_\Phi\, g_\Phi$

Figure 1: *Best-response dynamics* (BRD) algorithm based on our IPVI framework for DGPs.

the optimization of the ELBO as a two-player pure-strategy game between **Player 1** (representing discriminator with strategy $\{\Psi\}$) vs. **Player 2** (jointly representing generator and DGP model with strategy $\{\Phi, \theta\}$) that is defined based on the following payoffs:

$$
\begin{aligned}
\textbf{Player 1:} \quad &\max_{\{\Psi\}}\ \mathbb{E}_{p(\boldsymbol{\mathcal{U}})}[\log(1-\sigma(T_\Psi(\boldsymbol{\mathcal{U}})))] + \mathbb{E}_{q_\Phi(\boldsymbol{\mathcal{U}})}[\log\sigma(T_\Psi(\boldsymbol{\mathcal{U}}))]\,,\\
\textbf{Player 2:} \quad &\max_{\{\theta,\Phi\}}\ \mathbb{E}_{q_\Phi(\boldsymbol{\mathcal{U}})}[\mathcal{L}(\theta, \mathbf{X}, \mathbf{y}, \boldsymbol{\mathcal{U}}) - T_\Psi(\boldsymbol{\mathcal{U}})]\,.
\end{aligned}
\tag{7}
$$

**Proposition 2.** *Suppose that the parametric representations of $T_\Psi$ and $g_\Phi$ are expressive enough to represent any function. If $(\{\Psi^*\}, \{\theta^*, \Phi^*\})$ is a Nash equilibrium of the game in (7), then $\{\theta^*, \Phi^*\}$ is a global maximizer of the ELBO in (3) such that (a) $\theta^*$ is the maximum likelihood assignment for the DGP model, and (b) $q_{\Phi^*}(\boldsymbol{\mathcal{U}})$ is equal to the true posterior belief $p(\boldsymbol{\mathcal{U}}|\mathbf{y})$.*

Its proof is similar to that of Proposition 3 in [42] except that we additionally provide a proof of existence of a Nash equilibrium for the case of known/fixed DGP model hyperparameters, as detailed in Appendix B. Proposition 2 reveals that any Nash equilibrium coincides with a global maximizer of the original ELBO in (3). This consequently inspires us to play the game using *best-response dynamics*[3] (BRD) which is a commonly adopted procedure [2] to search for a Nash equilibrium. Fig. 1 illustrates our BRD algorithm: In each iteration of Algorithm 1, each player takes its turn to improve its strategy to achieve a better (but not necessarily the best) payoff by performing a *stochastic gradient ascent* (SGA) update on its payoff (7).

*Remark* 2. While BRD guarantees to converge to a Nash equilibrium in some classes of games (e.g., a finite potential game), we have not shown that our game falls into any of these classes and hence cannot guarantee that BRD converges to a Nash equilibrium (i.e., global maximizer $\{\theta^*, \Phi^*\}$) of our game. Nevertheless, as mentioned previously, obtaining the optimal discriminator in every iteration guarantees the game play (i.e., gradient ascent update for $\{\theta, \Phi\}$) to reach at least a local maximum of ELBO. To better approximate the optimal discriminator, we perform multiple calls of Algorithm 2 in every iteration of the main loop in Algorithm 1 and also apply a larger learning rate $\alpha_\Psi$. We have observed in our own experiments that these tricks improve the predictive performance of IPVI.

*Remark* 3. Existing implicit VI frameworks [52, 56] avoid the estimation of the log-density ratio. Unfortunately, the semi-implicit VI framework of [56] requires taking a limit at infinity to recover the ELBO, while the unbiased implicit VI framework of [52] relies on a Markov chain Monte Carlo sampler whose hyperparameters need to be carefully tuned.

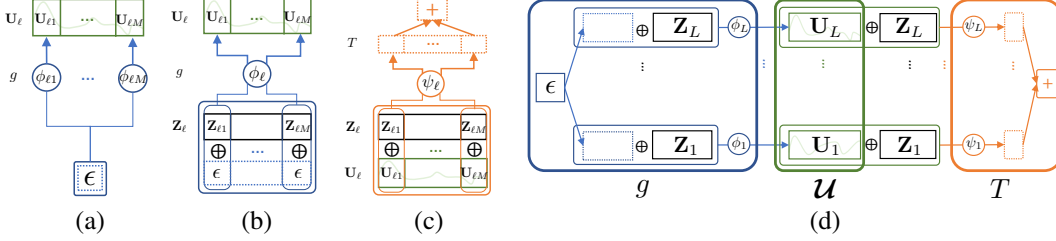

Figure 2: (a) Illustration of a naive design of the generator for each layer $\ell$. Parameter-tying architecture of the (b) generator and (c) discriminator for each layer $\ell$ where '+' denotes addition and '$\oplus$' denotes concatenation. (d) Parameter-tying architecture of the generator and discriminator in our IPVI framework for DGPs. See the main text for the definitions of notations.

# 4 Parameter-Tying Architecture of Generator and Discriminator for DGPs

In this section, we will discuss how the architecture of the generator in our IPVI framework is designed to enable parameter tying for a DGP model to alleviate overfitting. Recall from Section 2 that $\mathcal{U} = \{\mathbf{U}_\ell\}_{\ell=1}^L$ is a collection of inducing variables for DGP layers $\ell = 1, \ldots, L$. We consider a layer-wise design of the generator (parameterized by $\Phi \triangleq \{\phi_\ell\}_{\ell=1}^L$) and discriminator (parameterized by $\Psi \triangleq \{\psi_\ell\}_{\ell=1}^L$) such that $g_\Phi(\epsilon) \triangleq \{g_{\phi_\ell}(\epsilon)\}_{\ell=1}^L$ with the random noise $\epsilon$ serving as a common input to induce dependency between layers and $T_\Psi(\mathcal{U}) \triangleq \sum_{\ell=1}^L T_{\psi_\ell}(\mathbf{U}_\ell)$, respectively. For each layer $\ell$, a naive design is to generate posterior samples $\mathbf{U}_\ell \triangleq g_{\phi_\ell}(\epsilon)$ from the random noise $\epsilon$ as input. However, such a design suffers from two critical issues:

- Fig. 2a illustrates that to generate posterior samples of $M$ different inducing variables $\mathbf{U}_{\ell 1}, \ldots, \mathbf{U}_{\ell M}$ ($\mathbf{U}_\ell \triangleq \{\mathbf{U}_{\ell m}\}_{m=1}^M$), it is natural for the generator to adopt $M$ different parametric settings $\phi_{\ell 1}, \ldots, \phi_{\ell M}$ ($\phi_\ell \triangleq \{\phi_{\ell m}\}_{m=1}^M$), which introduces a relatively large number of parameters and is thus prone to overfitting (Section 5.3).
- Such a design of the generator fails to adequately capture the dependency of the inducing output variables $\mathbf{U}_\ell$ on the corresponding inducing inputs $\mathbf{Z}_\ell$, hence restricting its capability to output the posterior samples of $\mathcal{U}$ accurately.

To resolve the above issues, we propose a novel parameter-tying architecture of the generator and discriminator for a DGP model, as shown in Figs. 2b and 2c. For each layer $\ell$, since $\mathbf{U}_\ell$ depends on $\mathbf{Z}_\ell$, we design the generator $g_{\phi_\ell}$ to generate posterior samples $\mathbf{U}_\ell \triangleq g_{\phi_\ell}(\epsilon \oplus \mathbf{Z}_\ell)$ from not just $\epsilon$ but also $\mathbf{Z}_\ell$ as inputs. Recall that the same $\epsilon$ is fed as an input to $g_{\phi_l}$ in each layer $\ell$, which can be observed from the left-hand side of Fig. 2d. In addition, compared with the naive design in Fig. 2a, the posterior samples of $M$ different inducing variables $\mathbf{U}_{\ell 1}, \ldots, \mathbf{U}_{\ell M}$ are generated based on only a single shared parameter setting (instead of $M$), which reduces the number of parameters by $\mathcal{O}(M)$ times (Fig. 2b). We adopt a similar design for the discriminator, as shown in Fig. 2c. Fig. 2d illustrates the design of the overall parameter-tying architecture of the generator and discriminator.

We have observed in our own experiments that our proposed parameter-tying architecture not only speeds up the training and prediction, but also improves the predictive performance of IPVI considerably (Section 5.3). We will empirically evaluate our IPVI framework with this parameter-tying architecture in Section 5.

# 5 Experiments and Discussion

We empirically evaluate and compare the performance of our IPVI framework[4] against that of the state-of-the-art SGHMC [18] and *doubly stochastic VI*[5] (DSVI) [48] for DGPs based on their publicly

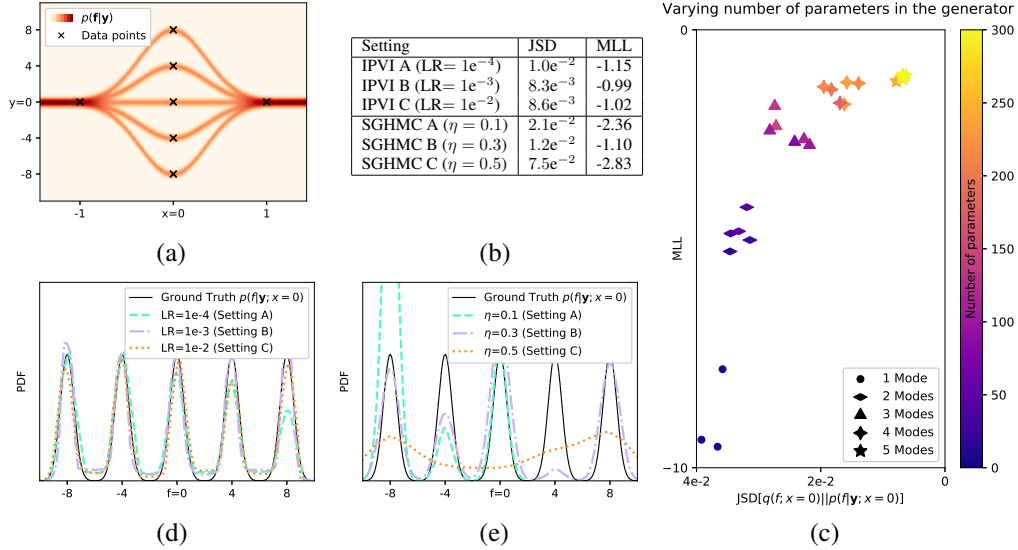

|  | (a) |  | (b) |  | (c) |
|---|---|---|---|---|---|

| Setting | JSD | MLL |
|---|---|---|
| IPVI A (LR= $1e^{-4}$) | $1.0e^{-2}$ | -1.15 |
| IPVI B (LR= $1e^{-3}$) | $8.3e^{-3}$ | -0.99 |
| IPVI C (LR= $1e^{-2}$) | $8.6e^{-3}$ | -1.02 |
| SGHMC A ($\eta = 0.1$) | $2.1e^{-2}$ | -2.36 |
| SGHMC B ($\eta = 0.3$) | $1.2e^{-2}$ | -1.10 |
| SGHMC C ($\eta = 0.5$) | $7.5e^{-2}$ | -2.83 |

(d)  (e)  (c)

Figure 3: (a) The *probability density function* (PDF) plot of the ground-truth posterior belief $p(\mathbf{f}|\mathbf{y})$. (b) Performances of IPVI and SGHMC in terms of estimated *Jenson-Shannon divergence* (JSD) and *mean log-likelihood* (MLL) metrics under the respective settings of varying learning rates $\alpha_\Psi$ and step sizes $\eta$. (c) Graph of MLL vs. JSD achieved by IPVI with varying number of parameters in the generator: Different shapes indicate varying number of modes learned by the generator. (d-e) PDF plots of variational posterior $q(f; x = 0)$ learned using (d) IPVI with generators of varying learning rates $\alpha_\Psi$ and (e) SGHMC with varying step sizes $\eta$.

available implementations using synthetic and real-world datasets in supervised (e.g., regression and classification) and unsupervised learning tasks.

## 5.1 Synthetic Experiment: Learning a Multi-Modal Posterior Belief

To demonstrate the capability of IPVI in learning a complex multi-modal posterior belief, we generate a synthetic "diamond" dataset and adopt a multi-modal mixture of Gaussian prior belief $p(\mathbf{f})$ (see Appendix C.1 for its description) to yield a multi-modal posterior belief $p(\mathbf{f}|\mathbf{y})$ for a single-layer GP. Fig. 3a illustrates this dataset and ground-truth posterior belief. Specifically, we focus on the multi-modal posterior belief $p(f|\mathbf{y}; x = 0)$ at input $x = 0$ whose ground truth is shown in Fig. 3d. Fig. 3c shows that as the number of parameters in the generator increases, the expressive power of IPVI increases such that its variational posterior $q(f; x = 0)$ can capture more modes in the true posterior, thus resulting in a closer estimated *Jensen-Shannon divergence* (JSD) between them and a higher *mean log-likelihood* (MLL).

Next, we compare the robustness of IPVI and SGHMC in learning the true multi-modal posterior belief $p(f|\mathbf{y}; x = 0)$ under different hyperparameter settings[6]: The generators in IPVI use the same architecture with about $300$ parameters but different learning rates $\alpha_\Psi$, while the SGHMC samplers use different step sizes $\eta$. The results in Figs. 3b and 3e have verified a remark made in [58] that SGHMC is sensitive to the step size which cannot be set automatically [49] and requires some prior knowledge to do so: Sampling with a small step size is prone to getting trapped in local modes while a slight increase of the step size may lead to an over-flattened posterior estimate. Additional results for different hyperparameter settings of SGHMC can be found in Appendix C.1. In contrast, the results in Figs. 3b and 3d reveal that, given enough parameters, IPVI performs robustly under a wide range of learning rates.

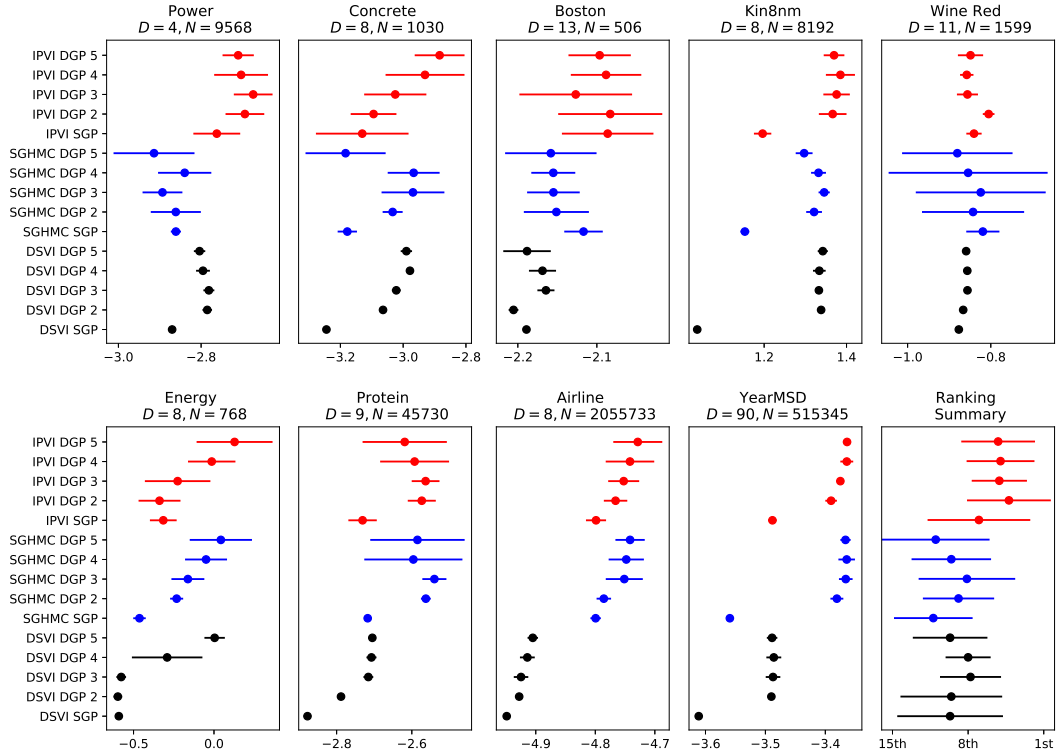

Figure 4: Test MLL and standard deviation achieved by our IPVI framework (red), SGHMC (blue), and DSVI (black) for DGPs for UCI benchmark and large-scale regression datasets. Higher test MLL (i.e., to the right) is better. See Appendix C.3 for a discussion on the performance gap between SGPs.

## 5.2 Supervised Learning: Regression and Classification

For our experiments in the regression tasks, the depth $L$ of the DGP models are varied from $1$ to $5$ with $128$ inducing inputs per layer. The dimension of each hidden DGP layer is set to be (i) the same as the input dimension for the UCI benchmark regression and Airline datasets, (ii) $16$ for the YearMSD dataset, and (iii) $98$ for the classification tasks. Additional details and results for our experiments (including that for IPVI with and without parameter tying) are found in Appendix C.3.

**UCI Benchmark Regression.** Our experiments are first conducted on 7 UCI benchmark regression datasets. We have performed a random $0.9/0.1$ train/test split.

**Large-Scale Regression.** We then evaluate the performance of IPVI on two real-world large-scale regression datasets: (i) YearMSD dataset with a large input dimension $D = 90$ and data size $N \approx 500000$, and (ii) Airline dataset with input dimension $D = 8$ and a large data size $N \approx 2$ million. For YearMSD dataset, we use the first $463715$ examples as training data and the last $51630$ examples as test data[7]. For Airline dataset, we set the last $100000$ examples as test data.

In the above regression tasks, the performance metric is the MLL of the test data (or test MLL). Fig. 4 shows results of the test MLL and standard deviation over $10$ runs. It can be observed that IPVI generally outperforms SGHMC and DSVI and the ranking summary shows that our IPVI framework for a 2-layer DGP model (IPVI DGP 2) performs the best on average across all regression tasks. For large-scale regression tasks, the performance of IPVI consistently increases with a greater depth. Even for a small dataset, the performance of IPVI improves up to a certain depth.

**Time Efficiency.** Table 1 and Fig. 5 show the better time efficiency of IPVI over the state-of-the-art SGHMC for a $4$-layer DGP model that is trained using the Airline dataset. The learning rates are $0.005$ and $0.02$ for IPVI and SGHMC (default setting adopted from [18]), respectively. Due to

Table 1: Time incurred by a 4-layer DGP model for Airline dataset.

| | IPVI | SGHMC |
|---|---|---|
| Average training time (per iter.) | 0.35 sec. | 3.18 sec. |
| $\mathcal{U}$ generation (100 samples) | 0.28 sec. | 143.7 sec. |

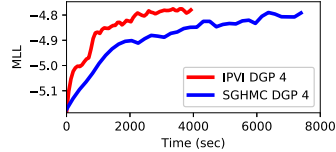

Figure 5: Graph of test MLL vs. total incurred time to train a 4-layer DGP model for the Airline dataset.

Table 2: Mean test accuracy (%) achieved by IPVI, SGHMC, and DSVI for 3 classification datasets.

| Dataset | MNIST | | MNIST ($M = 800$) | | Fashion-MNIST | | CIFAR-10 | |
|---|---|---|---|---|---|---|---|---|
| | SGP | DGP 4 | SGP | DGP 4 | SGP | DGP 4 | SGP | DGP 4 |
| DSVI | **97.32** | 97.41 | **97.92** | 98.05 | 86.98 | 87.99 | 47.15 | 51.79 |
| SGHMC | 96.41 | 97.55 | 97.07 | 97.91 | 85.84 | 87.08 | 47.32 | 52.81 |
| **IPVI** | 97.02 | **97.80** | 97.85 | **98.23** | **87.29** | **88.90** | **48.07** | **53.27** |

parallel sampling (Section 3) and a parameter-tying architecture (Section 4), our IPVI framework enables posterior samples to be generated 500 times faster. Although IPVI has more parameters than SGHMC, it runs 9 times faster during training due to efficiency in sample generation.

**Classification.** We evaluate the performance of IPVI in three classification tasks using the real-world MNIST, fashion-MNIST, and CIFAR-10 datasets. Both MNIST and fashion-MNIST datasets are grey-scale images of $28 \times 28$ pixels. The CIFAR-10 dataset consists of colored images of $32 \times 32$ pixels. We utilize a 4-layer DGP model with 100 inducing inputs per layer and a robust-max multiclass likelihood [21]; for MNIST dataset, we also consider utilizing a 4-layer DGP model with 800 inducing inputs per layer to assess if its performance improves with more inducing inputs. Table 2 reports the mean test accuracy over 10 runs, which shows that our IPVI framework for a 4-layer DGP model performs the best in all three datasets. Additional results for IPVI with and without parameter tying are found in Appendix C.3.

## 5.3 Parameter-Tying vs. No Parameter Tying

Table 3 reports the train/test MLL achieved by IPVI with and without parameter tying for 2 small datasets: Boston ($N = 506$) and Energy ($N = 768$). For Boston dataset, it can be observed that no tying consistently yields higher train MLL and lower test MLL, hence indicating overfitting. This is also observed for Energy dataset when the number of layers exceeds 2. For both datasets, as the number of layers (hence number of parameters) increases, overfitting becomes more severe for no tying. In contrast, parameter tying alleviates the overfitting considerably.

Table 3: Train/test MLL achieved by IPVI with and without parameter tying over 10 runs.

| Dataset | Boston ($N = 506$) | | | | |
|---|---|---|---|---|---|
| DGP Layers | 1 | 2 | 3 | 4 | 5 |
| No Tying | -1.86/-2.21 | -1.76/-2.37 | -1.64/-2.48 | -1.52/-2.51 | -1.51/-2.57 |
| Tying | -1.91/-2.09 | -1.79/-2.08 | -1.77/-2.13 | -1.84/-2.09 | -1.83/-2.10 |
| Dataset | Energy ($N = 768$) | | | | |
| DGP Layers | 1 | 2 | 3 | 4 | 5 |
| No Tying | -0.12/-0.44 | 0.03/-0.31 | 0.18/-0.34 | 0.20/-0.47 | 0.21/-0.58 |
| Tying | -0.16/-0.32 | -0.11/-0.34 | -0.02/-0.23 | 0.10/-0.01 | 0.17/ 0.13 |

## 5.4 Unsupervised Learning: FreyFace Reconstruction

A DGP can naturally be generalized to perform unsupervised learning. The representation of a dataset in a low-dimensional manifold can be learned in an unsupervised manner by the *GP latent variable model* (GPLVM) [33] where only the observations $\mathbf{Y} \triangleq \{\mathbf{y}_n\}_{n=1}^N$ are given and the hidden representation $\mathbf{X}$ is unobserved and treated as latent variables. The objective is to infer the posterior

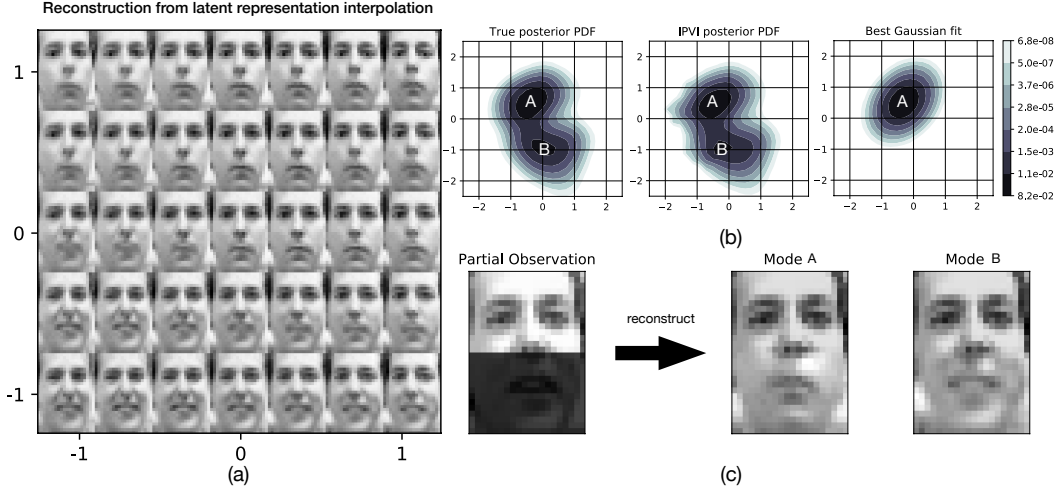

Figure 6: Unsupervised learning with FreyFace dataset. (a) Latent representation interpolation and the corresponding reconstruction. (b) True posterior $p(\mathbf{x}^\star|\mathbf{y}_O^\star)$ given the partial observation $\mathbf{y}_O^\star$ (left), variational posterior $q(\mathbf{x}^\star)$ learned by IPVI (middle), and Gaussian approximation (right). The PDF for $p(\mathbf{x}^\star|\mathbf{y}_O^\star)$ is calculated using Bayes rule where the marginal likelihood is computed using Monte Carlo integration. (c) The partial observation (with the ground truth reflected in the dark region) and two reconstructed samples from $q(\mathbf{x}^\star)$.

$p(\mathbf{X}|\mathbf{Y})$. The GPLVM is a single-layer GP that casts $\mathbf{X}$ as an unknown distribution and can naturally be extended to a DGP. So, we construct a 2-layer DGP ($\mathbf{X} \rightarrow \mathbf{F}_1 \rightarrow \mathbf{F}_2 \rightarrow \mathbf{Y}$) and use the generator samples to represent $p(\mathbf{X}|\mathbf{Y})$.

We consider the FreyFace dataset [47] taken from a video sequence that consists of 1965 images with a size of $28 \times 20$. We select the first 1000 images to train our DGP. To ease visualization, the dimension of latent variables $\mathbf{X}$ is chosen to be 2. Additional details for our experiments are found in Appendix C.2. Fig. 6a shows the reconstruction of faces across the latent space. Interestingly, the first dimension of the latent variables $\mathbf{X}$ determines the expression from happy to calm while the the second dimension controls the view angle of the face.

We then explore the capability of IPVI in reconstructing partially observed test data. Fig. 6b illustrates that given only the upper half of the face, the real face may exhibit a multi-modal property, as reflected in the latent space; intuitively, one cannot always tell whether a person is happy or sad by looking at the upper half of the face. Our variational posterior accurately captures the multi-modal posterior belief whereas the Gaussian approximation can only recover one mode (mode A) under this test scenario. So, IPVI can correctly recover two types of expressions: calm (mode A) and happy (mode B). We did not empirically compare with SGHMC here because it is not obvious to us whether their sampler setting can be carried over to this unsupervised learning task.

## 6 Conclusion

This paper describes a novel IPVI framework for DGPs that can ideally recover an unbiased posterior belief of the inducing variables and still preserve the time efficiency of VI. To achieve this, we cast the DGP inference problem as a two-player game and search for a Nash equilibrium (i.e., an unbiased posterior belief) of this game using best-response dynamics. We propose a novel parameter-tying architecture of the generator and discriminator in our IPVI framework for DGPs to alleviate overfitting and speed up training and prediction. Empirical evaluation shows that IPVI outperforms the state-of-the-art approximation methods for DGPs in regression and classification tasks and accurately learns complex multi-modal posterior beliefs in our synthetic experiment and an unsupervised learning task. For future work, we plan to use our IPVI framework for DGPs to accurately represent the belief of the unknown target function in active learning [4, 28, 35, 37–39, 44, 60] and Bayesian optimization [11, 13, 26, 34, 59, 61] when the available budget of function evaluations is moderately large. We also plan to develop distributed/decentralized variants [5–8, 23, 25, 27, 40, 43] of IPVI.

**Acknowledgements.** This research is supported by the National Research Foundation, Prime Minister's Office, Singapore under its Campus for Research Excellence and Technological Enterprise (CREATE) program, Singapore-MIT Alliance for Research and Technology (SMART) Future Urban Mobility (FM) IRG, National Research Foundation Singapore under its AI Singapore Programme Award No. AISG-GC-2019-002, and the Singapore Ministry of Education Academic Research Fund Tier 2, MOE2016-T2-2-156.

## Footnotes

[2]An alternative is to modify the DGP prior directly and perform inference with a parametric model. The work of [9] has approximated the DGP prior with the spectral density of a kernel [22] such that the kernel has an analytical spectral density.

[3]This procedure is sometimes called "better-response dynamics" (http://timroughgarden.org/f13/l/l16.pdf).

[4]Our implementation is built on GPflow [41] which is an open-source GP framework based on TensorFlow [1]. It is publicly available at https://github.com/HeroKillerEver/ipvi-dgp.

[5]It is reported in [48] that DSVI has outperformed the approximate expectation propagation method of [3] for DGPs. Hence, we do not empirically compare with the latter [3] here.

[6]We adopt scale-adapted SGHMC which is a robust variant used in Bayesian neural networks and DGP inference [18]. A recent work of [58] has proposed the cyclical stochastic gradient MCMC method to improve the accuracy of sampling highly complex distributions. However, it is not obvious to us how this method can be incorporated into DGP models, which is beyond the scope of this work.

[7]This avoids the 'producer' effect by ensuring that no song from an artist appears in both training & test data.

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
