[Supplementary Material · neurips2019_camera_ready_app.pdf]

# A    Proof of Proposition 1

The objective function in (4) can be re-written as

$$\int p(\boldsymbol{\mathcal{U}}) \log(1 - \sigma(T(\boldsymbol{\mathcal{U}}))) \, \mathrm{d}\boldsymbol{\mathcal{U}} + \int q_\Phi(\boldsymbol{\mathcal{U}}) \log \sigma(T(\boldsymbol{\mathcal{U}})) \, \mathrm{d}\boldsymbol{\mathcal{U}} \, .$$

The above integral is maximal in function $T$ if and only if the integrand is maximal in $T(\boldsymbol{\mathcal{U}})$ for every $\boldsymbol{\mathcal{U}}$. Note that the maximum of $a \log(t) + b \log(1 - t)$ over $t \in [0, 1]$ is at $t = a/(a + b)$ for any $(a, b) \in \mathbb{R}^2 \backslash (0, 0)$. Using this result,

$$\sigma(T^*(\boldsymbol{\mathcal{U}})) = \frac{q_\Phi(\boldsymbol{\mathcal{U}})}{q_\Phi(\boldsymbol{\mathcal{U}}) + p(\boldsymbol{\mathcal{U}})}$$

or, equivalently,

$$T^*(\boldsymbol{\mathcal{U}}) = \log q_\Phi(\boldsymbol{\mathcal{U}}) - \log p(\boldsymbol{\mathcal{U}}) \, .$$

# B    Proof of Proposition 2

If $(\{\Psi^*\}, \{\theta^*, \Phi^*\})$ is a Nash equilibrium, then according to Proposition 1 and under the assumption that $T_{\Psi^*}$ is expressive enough, we know that **Player 1** is playing its optimal strategy $\Psi^*$ such that

$$T_{\Psi^*}(\boldsymbol{\mathcal{U}}) = \log q_{\Phi^*}(\boldsymbol{\mathcal{U}}) - \log p(\boldsymbol{\mathcal{U}}) \, . \tag{8}$$

Substituting (8) into (6) reveals that **Player 2**'s strategy $\{\theta^*, \Phi^*\}$ maximizes its payoff which is a function of $\{\theta, \Phi\}$:

$$\begin{aligned}
\mathcal{F}(\theta, \Phi) &\triangleq \mathbb{E}_{q_\Phi(\boldsymbol{\mathcal{U}})}[\mathcal{L}(\theta, \mathbf{X}, \mathbf{y}, \boldsymbol{\mathcal{U}}) + \log p(\boldsymbol{\mathcal{U}}) - \log q_{\Phi^*}(\boldsymbol{\mathcal{U}})] \\
&= \mathbb{E}_{q_\Phi(\boldsymbol{\mathcal{U}})}[\mathcal{L}(\theta, \mathbf{X}, \mathbf{y}, \boldsymbol{\mathcal{U}}) + \log p(\boldsymbol{\mathcal{U}}) - \log q_\Phi(\boldsymbol{\mathcal{U}}) + \log q_\Phi(\boldsymbol{\mathcal{U}}) - \log q_{\Phi^*}(\boldsymbol{\mathcal{U}})] \\
&= \mathcal{EL}(\theta, \Phi) + \mathrm{KL}[q_\Phi(\boldsymbol{\mathcal{U}}) \| q_{\Phi^*}(\boldsymbol{\mathcal{U}})]
\end{aligned} \tag{9}$$

where $\mathcal{EL}(\theta, \Phi)$ is the ELBO in (3).

Now, suppose that $\{\theta^*, \Phi^*\}$ does not maximize the ELBO. Then, there exists some $\{\theta', \Phi'\}$ such that $\mathcal{EL}(\theta', \Phi') > \mathcal{EL}(\theta^*, \Phi^*)$. By substituting $\{\theta', \Phi'\}$ into (9),

$$\mathcal{F}(\theta', \Phi') = \mathcal{EL}(\theta', \Phi') + \mathrm{KL}[q_{\Phi'}(\boldsymbol{\mathcal{U}}) \| q_{\Phi^*}(\boldsymbol{\mathcal{U}})] > \mathcal{F}(\theta^*, \Phi^*) \, ,$$

which contradicts the fact that $\{\theta^*, \Phi^*\}$ maximizes (9). Hence, $\{\theta^*, \Phi^*\}$ maximizes the ELBO, which is equal to the log-marginal likelihood $\log p_{\theta^*}(\mathbf{y})$ with $\theta^*$ being the maximum likelihood assignment and $q_{\Phi^*}(\boldsymbol{\mathcal{U}})$ being equal to the true posterior belief $p(\boldsymbol{\mathcal{U}}|\mathbf{y})$.

## B.1    Discussion on the Existence of Nash Equilibrium

**Proposition 3.** *Suppose that the parametric representations of $T_\Psi$ and $g_\Phi$ are expressive enough to represent any function and the DGP model hyperparameters are fixed to be $\theta_\circ$. Then, the two-player pure-strategy game in (7) for the case of fixed $\theta_\circ$ has a Nash equilibrium. Furthermore, if $(\{\Psi^*\}, \{\theta_\circ, \Phi^*\})$ is a Nash equilibrium, then $\{\Phi^*\}$ is a global maximizer of the ELBO for the case of fixed $\theta_\circ$ such that $q_{\Phi^*}(\boldsymbol{\mathcal{U}})$ is equal to the true posterior belief $p_{\theta_\circ}(\boldsymbol{\mathcal{U}}|\mathbf{y})$.*

*Proof.* Since we assume the parametric representation of $g_\Phi$ to be expressive enough to represent any function, we can find some $\{\Phi_\circ\}$ such that $q_{\Phi_\circ}(\boldsymbol{\mathcal{U}})$ is equal to the true posterior belief $p_{\theta_\circ}(\boldsymbol{\mathcal{U}}|\mathbf{y})$. We now know that $\{\Phi_\circ\}$ maximizes the ELBO in (3) for the case of fixed DGP model hyperparameters $\theta_\circ$, which we denote by $\mathcal{EL}(\theta_\circ, \Phi_\circ)$.

Since we assume the parametric representation of $T_\Psi$ to be expressive enough to represent any function, we can further obtain some $\{\Psi_\circ\}$ such that $T_{\Psi_\circ}(\boldsymbol{\mathcal{U}}) = \log q_{\Phi_\circ}(\boldsymbol{\mathcal{U}}) - \log p(\boldsymbol{\mathcal{U}})$. According to Proposition 1, $\{\Psi_\circ\}$ maximizes the payoff to **player 1**. Hence, **player 1** cannot improve its strategy to achieve a better payoff.

Given that **player 1** plays strategy $\{\Psi_\circ\}$ for the case of fixed $\theta_\circ$, the payoff to **player 2** playing strategy $\{\theta_\circ, \Phi\}$ is

$$
\begin{aligned}
\mathcal{F}(\theta_\circ, \Phi) &\triangleq \mathbb{E}_{q_\Phi(\mathcal{U})}[\mathcal{L}(\theta_\circ, \mathbf{X}, \mathbf{y}, \mathcal{U}) + \log p(\mathcal{U}) - \log q_{\Phi_\circ}(\mathcal{U})] \\
&= \mathbb{E}_{q_\Phi(\mathcal{U})}[\mathcal{L}(\theta_\circ, \mathbf{X}, \mathbf{y}, \mathcal{U}) + \log p(\mathcal{U}) - \log q_\Phi(\mathcal{U}) + \log q_\Phi(\mathcal{U}) - \log q_{\Phi_\circ}(\mathcal{U})] \\
&= \mathcal{EL}(\theta_\circ, \Phi) + \mathrm{KL}[q_\Phi(\mathcal{U}) \| q_{\Phi_\circ}(\mathcal{U})] \\
&= \log p_{\theta_\circ}(\mathbf{y}) - \mathrm{KL}[q_\Phi(\mathcal{U}) \| p_{\theta_\circ}(\mathcal{U}|\mathbf{y})] + \mathrm{KL}[q_\Phi(\mathcal{U}) \| q_{\Phi_\circ}(\mathcal{U})] \\
&= \log p_{\theta_\circ}(\mathbf{y}) \ .
\end{aligned}
$$

So, **player 2** receives a constant payoff (i.e., independent of $\{\Phi, \theta_\circ\}$) and cannot improve its strategy to achieve a better payoff. Since every player cannot improve strategy to achieve a better payoff, $(\{\Psi_\circ\}, \{\theta_\circ, \Phi_\circ\})$ is a Nash Equilibrium.

The rest of the proof is similar to that of Proposition 2. $\qquad\square$

Given that the hyperparameters $\theta_\circ$ of a single-layer DGP (i.e., SGP) regression model are fixed, the true posterior belief $p_{\theta_\circ}(\mathcal{U}|\mathbf{y})$ is guaranteed to be a Gaussian [51]. In this case, Proposition 3 indicates that $q_{\Phi^*}(\mathcal{U})$ is equal to this Gaussian.

# C  Additional Details for Experiments

## C.1  Synthetic Experiment: Learning a Multi-Modal Posterior Belief

The prior belief is set as a mixture of 5 Gaussians:

$$
p(\mathbf{f}) \triangleq p_i \sum_{i=1}^{5} \mathcal{N}(\mu_i \exp(-8x^2), \mathbf{K_{XX}})
$$

where $p_i \triangleq 1/5$ for $i = 1, \ldots, 5$, $\mu_1 \triangleq -8$, $\mu_2 \triangleq -4$, $\mu_3 \triangleq 0$, $\mu_4 \triangleq 4$, $\mu_5 \triangleq 8$, and $\mathbf{K_{XX}}$ denotes a constant covariance matrix with a constant kernel $k(x, x') \triangleq \sigma_A^2$ and $\sigma_A^2 \triangleq 1/(4 - \exp(-8))$.

Also, $p(\mathbf{y}|\mathbf{f}) = \prod_n p(y_n|f_n) = \prod_n (1/(\sqrt{2\pi}\sigma_B)) \exp(-(y_i - f_i)^2/(2\sigma_B^2))$ with a large noise variance $\sigma_B^2 = 7\exp(8)$. Then, the ground-truth posterior belief with 5 modes can be recovered analytically using Bayes rule:

$$
p(\mathbf{f}|\mathbf{y}) = p_i' \sum_{i=1}^{5} \mathcal{N}(\mu_i \exp(-8x^2) + \delta_i, \mathbf{K'_{XX}})
$$

where $p_1' = 0.1988$, $p_2' = 0.2004$, $p_3' = 0.2016$, $p_4' = 0.2004$, $p_5' = 0.1988$, $\delta_1 = 0.000479$, $\delta_2 = 0.00024$, $\delta_3 = 0$, $\delta_4 = -0.00024$, $\delta_5 = -0.000479$, and $\mathbf{K'_{XX}}$ denotes a constant covariance matrix with a constant kernel $k'(x, x') \triangleq \sigma_C^2$ and $\sigma_C^2 = 1/4$.

In our implementation, the ground-truth GP kernel hyperparameter values are known to IPVI and SGHMC. We adopt a single inducing input fixed at $z = 0$. The multi-modal posterior belief $p(f|\mathbf{y}; x = 0)$ is then approximated using the samples from $p(u|\mathbf{y}; z = 0)$. In Fig. 7, we give additional results for different hyperparameter settings of SGHMC to show that it is likely to obtain a biased posterior belief.

We vary the number of hidden layers and number of neurons in each hidden layer to obtain generators with different number of parameters in Fig. 3c.

## C.2  Unsupervised Learning: FreyFace Reconstruction

The dimensions of the hidden layers are 2 for $\mathbf{X}$ and 100 for $\mathbf{F}_1$ for FreyFace Reconstruction. We did not exploit inducing variables here. So, the training is a full DGP. We use PCA as the mean function for this unsupervised learning task.

**Reconstruction.** Given a trained DGP model, the reconstruction task of a partially observed $\mathbf{y}_O^\star$ is to recover the missing part $\mathbf{y}_U^\star$ such that $\mathbf{y}^\star = [\mathbf{y}_O^\star, \mathbf{y}_U^\star]$. This reconstruction task involves two steps. The first step is to cast it as an DGP inference problem to get the posterior $p(\mathbf{x}^\star|\mathbf{y}_O^\star)$ with a Gaussian likelihood $p(\mathbf{y}_O^\star|\mathbf{y}^\star)$. The second step samples $\mathbf{y}^\star$ from $p(\mathbf{y}^\star|\mathbf{y}_O^\star) = \int p(\mathbf{y}^\star|\mathbf{x}^\star)\, p(\mathbf{x}^\star|\mathbf{y}_O^\star)\, \mathrm{d}\mathbf{x}^\star$.

Figure 7: SGHMC with different hyperparameter settings of learning rate $\eta$, momentum $1 - \alpha$, Fisher information $V$, and initialization init for starting the sampler: (a) $\eta = 0.3, \alpha = 0.4, V = 0.1$; (b) $\eta = 0.3, \text{init} = 4, \alpha = 0.4$; and (c) $\eta = 0.3, \text{init} = 4, V = 0.1$.

## C.3 Supervised Learning: Regression and Classification

In this subsection, we provide additional details for our experiments in the supervised learning tasks.

**Learning Rates.** We adopt the default settings of the learning rates of the tested methods from their publicly available implementations. The learning rates and maximum iteration for IPVI are tuned through grid search and cross validation with a default setting of $\alpha_\Psi = 0.05$, $\alpha_\Phi = 0.001$, $\alpha_\theta = 0.025$ and cut-off at a maximum of 20000 iterations. The learning rates for classification is simply set to be 0.02 for all parameters.

**Hidden Dimensions.** The dimension of inducing variables for all implementations are set to be (i) the same as input dimension for the UCI benchmark regression and Airline datasets, (ii) 16 for the YearMSD dataset, and (iii) 98 for the classification tasks.

**Mini-Batch Sizes.** The mini-batch sizes for all implementations are set to be (i) 10000 for the UCI benchmark regression tasks, (ii) 20000 for the large-scale regression tasks, and (iii) 256 for the classification tasks.

**Generator/Discriminator Details.** We have described the architecture design in Section 4. We will describe here the neural network represented by $g_{\phi_\ell}$. Firstly, the noise $\epsilon$ has the same dimension as the inputs $\mathbf{X}$ of the dataset. We implement $g_{\phi_\ell}$ using a two-layer neural network with hidden dimension being equal to the dimension of $\mathbf{Z}_\ell$ and leaky ReLU activation in the middle. Similarly, we implement $T_{\psi_\ell}$ using a two-layer neural network with hidden dimension being equal to the dimension of $\mathbf{Z}_\ell$ and leaky ReLU activation in the middle. The network initialization follows random normal distribution.

**Mean Function of DGP.** The 'skip-layer' connections are implemented in both SGHMC [18] and DSVI [48] for DGPs and in our IPVI framework as well. The work of [15] has analyzed that using a zero mean function in the DGP prior causes some difficulty as each GP mapping is highly non-injective. To mitigate this issue, the work of [48] has proposed to include a linear mean function $m(\mathbf{X}) = \mathbf{W}\mathbf{X}$ for all hidden layers. The 'skip-layer' connection $\mathbf{W}$ is set to be an identity matrix if the input dimension equals to the output dimension. Otherwise, $\mathbf{W}$ is computed from the top $H$ eigenvectors of the data under SVD. We follow the same setting as this 'skip-layer' mean function. Note that this 'skip-layer' mean function contains no trainable parameters.

**Likelihood.** For the classification tasks, we use the robust-max multiclass likelihood [21]. Tricks like data augmentation are not applied, which means that the accuracy can still be improved further with those additional tricks.

**Parameter-Tying vs. No Parameter-Tying.** Tables 4 and 5 show, respectively, results of the test MLL for more UCI benchmark regression datasets and the mean test accuracy for the three classification tasks over 10 runs that are achieved by IPVI with and without parameter tying. It can be observed that IPVI achieves a considerably better predictive performance with parameter tying.

**Performance Gap between SGPs.** Regarding the performance gap between SGPs, note that the optimal variational posterior is a Gaussian for a SGP regression model [51]. However, since the SGP model hyperparameters are not known beforehand, DSVI SGP has to jointly optimize its hyperparameters and variational parameters. Such an optimization is not convex. Hence, there is

Table 4: Test MLL achieved by our IPVI framework with and without parameter tying for UCI benchmark regression datasets. Higher test MLL is better.

| Dataset | Boston | | | | | Power | | | | |
|---|---|---|---|---|---|---|---|---|---|---|
| DGP Layers | 1 | 2 | 3 | 4 | 5 | 1 | 2 | 3 | 4 | 5 |
| No Tying | -2.21 | -2.37 | -2.48 | -2.51 | -2.57 | -2.77 | -2.79 | -2.74 | -2.73 | -2.75 |
| Tying | -2.09 | -2.08 | -2.13 | -2.09 | -2.10 | -2.76 | -2.69 | -2.67 | -2.70 | -2.71 |
| Dataset | Wine Red | | | | | Protein | | | | |
| DGP Layers | 1 | 2 | 3 | 4 | 5 | 1 | 2 | 3 | 4 | 5 |
| No Tying | -0.97 | -0.94 | -0.96 | -0.97 | -0.97 | -2.83 | -2.72 | -2.69 | -2.70 | -2.67 |
| Tying | -0.84 | -0.81 | -0.86 | -0.86 | -0.85 | -2.73 | -2.57 | -2.56 | -2.59 | -2.62 |

Table 5: Mean test accuracy (%) achieved by our IPVI framework with and without parameter tying for three classification datasets.

| Dataset | MNIST | | fashion-MNIST | | CIFAR-10 | |
|---|---|---|---|---|---|---|
| DGP Layers | 1 | 4 | 1 | 4 | 1 | 4 |
| No Tying | 96.77 | 97.45 | 86.69 | 88.01 | 47.13 | 52.76 |
| Tying | 97.02 | 97.80 | 87.29 | 88.90 | 48.07 | 53.27 |

no guarantee that it will reach the global optimum. Thus, the performance gap can be explained by IPVI's ability to jointly find "better" values of hyperparameters and variational parameters.

**Evaluation of ELBO.** We have also computed the estimate of ELBO by, after training our IPVI DGP models for the Boston dataset, continuing to train the discriminator using more calls of Algorithm 2. Table 6 shows the mean ELBOs of DSVI and IPVI over 10 runs for the Boston dataset. IPVI generally achieves higher ELBOs, which agrees with results of the test MLL in Fig. 4. Since SGHMC DGP is not based on VI, no ELBO is computed for that method.

Table 6: Mean ELBOs for Boston dataset.

| Model | DSVI | IPVI |
|---|---|---|
| SGP | -956.57 | -934.07 |
| DGP 2 | -850.54 | -846.65 |
| DGP 3 | -836.13 | -846.45 |
| DGP 4 | -787.10 | -776.93 |
| DGP 5 | -770.67 | -758.42 |