[Reviews · NeurIPS 2019]

Reviewer 1



To me, this is an important paper contributing significantly to Bayesian deep leaning. Specifically, the paper bring the idea of “adversarial variational Bayes” to deep Gaussian processes, which is both novel (although someone may argue the idea already appears in variational autoencoders) and important. As pointed out by the authors, the learning of DGP is significantly harder than a shallow GP, even after introducing sparse approximation, and the field is dominated by the mean-field variational inference (which is easy to implement and works robustly in practice, but may lose predictive powers due to the mean-field assumption) and more recently stochastic MCMC such as SGHMC (which promises better results but is hard to tune in practice). All these urge us to bring new and better methods to training DGP or even general Bayesian deep learning models (such as Bayesian neural networks). The idea of “adversarial variational Bayes” or “implicit posterior” is a promising direction to go and the work in this paper demonstrates a significant step. In my opinion, the paper makes the following contributions: (1) The paper introduces the novel idea of “implicit posterior variational inference” to DGP, the methodology is introduced clearly and the method is solid, both theoretically and empirically; (2) The authors introduced some modifications of the architecture tailoring to the problem of DGP itself, such as parameter tying, concatenating the random inputs and the inducing points (and inducing variables); (3) Both theoretical results and the experimental section are complete and convincing. A structure suggestion hopefully to make the paper even better: Although I understand for completeness, the authors give pretty detailed explanations to “implicit posterior variational inference” in Sect 3, however, the parts have some overlaps with [1], for example the idea of introducing T (discriminator) to compute the KL divergence, etc. In contrast, Sect 4 is pretty novel and important for the method to work in DGP, while is quite short. I suggest to shorten Sect 3 a bit (by making more connections to [1], and/or move some parts to the appendix) and give more details in Sect 4 (such as a demo how parameter tying helps comparing to a naïve design). Minor comments: From Section 4, it’s not clear to me epsilon (random inputs) are shared among all the layers or not. [1] Mescheder, Lars, Sebastian Nowozin, and Andreas Geiger. "Adversarial variational bayes: Unifying variational autoencoders and generative adversarial networks." Proceedings of the 34th International Conference on Machine Learning-Volume 70. JMLR. org, 2017.

Reviewer 2



(1) Quality: Is the submission technically sound? Are claims well supported by theoretical analysis or experimental results? Is this a complete piece of work or work in progress? Are the authors careful and honest about evaluating both the strengths and weaknesses of their work? The paper is good quality. The idea of using IPVI for DGPs is interesting and it seems to perform competitively with VI and SGHMC. Experiments are conducted on both toy data, for demonstration purposes, and the most common benchmark datasets. One aspect that this work could be improved in is further comparison and discussion on the design choices of the IPVI algorithm. Despite the wealth of research on implicit VI (Semi-Implicit Variational Inference by Yin and Zhou, Unbiased Implicit Variational Inference by Titsias and Ruiz), only (Huszar 2017) is cited and briefly discussed. I believe that the use of IPVI warrants a more in-depth discussion. (2) Clarity: Is the submission clearly written? Is it well organized? (If not, please make constructive suggestions for improving its clarity.) Does it adequately inform the reader? (Note: a superbly written paper provides enough information for an expert reader to reproduce its results.) The paper is well organized and easy to read. It provides good background on DGPs and VI. The IPVI algorithm is described in detail, including pseudocode. It would be possible to reproduce the results based on the paper. (3) Originality: Are the tasks or methods new? Is the work a novel combination of well-known techniques? Is it clear how this work differs from previous contributions? Is related work adequately cited? The application of IPVI to DGPs is certainly new and interesting, and I like the idea of parameter tying. However, Implicit VI in itself is not new, and this should be made clear. I would recommend citing (Huszar, 2017) in the introduction to make sure that the reader gets the right impression. (4) Significance: Are the results important? Are others (researchers or practitioners) likely to use the ideas or build on them? Does the submission address a difficult task in a better way than previous work? Does it advance the state of the art in a demonstrable way? Does it provide unique data, unique conclusions about existing data, or a unique theoretical or experimental approach? I believe that the work is significant to practitioners, as it is an effective way to do inference in DGPs. ___________________________________________________________________________________________- After reading the author's reply, I decided to keep my original score.

Reviewer 3



The constraints placed on approximate posteriors for deep Gaussian processes have long been a problem. This paper eliminates this problem by allowing implicitly defined distributions over the inducing outputs u, which are very flexible. The drawback is that the density ratio, which is required for the KL, can not be evaluated any more. However, this density ratio can be estimated using an adversarial procedure. This creates a GAN-like training procedure, where two conflicting objective functions are simultaneously optimised. Technically, this paper is sound, and the experimental results are clearly encouraging. The method, as applied to the deep GP model, is novel, and appropriate. Taken together, I believe this is a strong case for accepting the paper. However, there are several points which need clarification if I am to argue for acceptance of this paper. 1. The necessity of parameter tying Experimentally, the paper shows that parameter tying is needed for improved performance. However, the explanation given in the paper in §4 does not fit with the theoretical properties of variational inference. Firstly, regarding the generator. The flexibility of the generator determines the range of variational posteriors q(u) that are possible. Hence, the generator's parameters are variational parameters, which are trained to come close to the true posterior, rather that fit the data. This causes the generator's parameters to be inherently protected from overfitting. This is the most common justification for using variational inference in the first place. I therefore find it strange that suddenly overfitting is used to explain diminished performance, without any experimental evidence except diminished performance. Optimization difficulty seems like a more reasonable explanation to me. Either way, experimental evidence is needed, or the observation should be stated as unexplained. A similar argument goes for the discriminator for the KL. In principle, the more flexible this discriminator, the more accurate the density ratio estimate can be. These parameters are not fit to the data either, and so overfitting is not a likely solution. 2. Depth of experimental investigation Experiments should be used to convey insights into methods, i.e. why they perform in a particular way, in addition to simply benchmarking. The following questions are unresolved: - Single layer GPs (SGP vs IPVI SGP) It is a good idea to include the single layer GP in the comparison. It is surprising, however, that the IPVI SGP performs better than the DSVI GP. Given the phrasing of the paper, it is implied that the difference in performance should be explained by the difference in flexibility of the variational distribution q(u). However, in the regression experiments (where, presumably a Gaussian likelihood is used), the optimal q(u) is Gaussian! The difference can therefore not be explained by the more flexible variational distribution! Is it simply that DSVI SGP trains slower? I.e. is performance is limited by the 20000 iterations? You could run Titsias' 2009 method, which computes q(u) in closed form, to be sure. Given this discrepancy, this may also cast doubts on the mechanism of the improvement for other experiments. I believe that the model assumptions, as well as poor quality of inference holds back the performance of deep GP models. Perhaps this method performs worse inference, but allows the data to be fit more closely in contradiction to model assumptions? The toy example does give a (somewhat synthetic) example of the flexibility of the inference, but more investigation in the more practical cases would be appreciated. Can we compute the estimate of the ELBO for the new method? For example by, after training, continuing to train the discriminator for longer, and then to evaluate a stochastic estimate of the ELBO. This can then be compared to the ELBO of other methods. A higher value would provide evidence in favour of better inference. Finally, for the MNIST example, the number of inducing points is far too small. Results of 1.9% error have been reported with a single layer GP with an RBF kernel if when 750 inducing points are used. Constraining the number of inducing points makes the inference poor, and is not the interesting regime for comparison. The code looks of high quality, but bears a striking resemblance to the open source package GPflow (although edited in certain ways), even down to function and variable names. If the code was based on GPflow, a citation should be considered. Overall, a very interesting paper, which addresses an important problem in variational inference for deep GPs. ===== Response to rebuttal Many thanks for the rebuttal. I will argue for acceptance of this paper and have increased my score accordingly, even though I do not agree with all the resolutions the authors provide. - VI and overfitting I agree with the authors that VI can still be subject to overfitting, but I strongly disagree with the understanding and reasoning that the authors provide. The crux of our disagreement is in the authors' belief that overfitting in VI becomes worse as more variational parameters are added. This is the opposite of what happens. In fact, VI has the potential to overfit more the more the variational family is constrained! The classic example of this is that if you constrain the variational family to be mean-field (i.e. little flexibility!) then you under-estimate uncertainties [1, 2]. The core assumption I am making, is that the authors and I agree that the true posterior does not overfit. The more flexibility (approximately measured by number of parameters) you add to the variational family, the closer you can get to the true posterior, and the less the solution will overfit. Note that I am not arguing with the empirical findings of the authors, simply with the explanation they give for them! To me, if adding parameters to the variational approximation leads to diminished performance, a different explanation can be found. Possible explanations are: - More difficult optimisation leads to a poorer approximate posterior to be found, despite the variational family being larger. - The approximation to variational inference that the paper introduces is failing (i.e. the discriminator is not accurate). - The model is misspecified and so deviating more from the true posterior leads to improved performance. It is crucial for the progress in Bayesian machine learning approaches that these issues are investigated thoroughly, and that the reasons for unexpected behaviour are unpicked carefully. The advantage of the Bayesian approach is that it separates optimisation issues, modelling issues, and overfitting issues. Calling something an overfitting issue when it is something else is unhelpful to understanding the underlying problem, and does not do justice to the ability of the Bayesian framework to pick these issues apart. Conflation of issues surrounding optimisation, approximation and overfitting have happened in the past in the GP community, as discussed in e.g. [3]. - Joint optimisation of hyperparameters and variational parameters The optimisation of q(u) in SGP (for fixed hyperparameters) is convex. Since the hyperparameters are guaranteed to converge to a local optimum, we also have a guarantee that q(u) will converge to the optimum for those hyperparameters. The non-convexity of the optimisation in the hyperparameters has never been a problem for me for the UCI datasets. If that were the case, I would have observed more variation in performance for different random restarts. Regardless, there is still a discrepancy here that should be investigated. To confirm correct working of your approximation to variational inference, you could initialise an SGP model to the hyperparameters obtained from your method, and then find q(u) in closed form [4]. Any difference in performance in this case would be explained by your method deviating from variational inference. This would imply that your method is not strictly doing variational inference. This is not a grounds for rejection, but only an important observation that would only strengthen the evaluation of the work! I really appreciate the tables of ELBO estimates, and the MNIST run with more inducing points. These results are much more in line with what I expect [1] http://www.inference.org.uk/itprnn/book.pdf fig 33.6 page 436 [2] http://www.gatsby.ucl.ac.uk/~maneesh/papers/turner-sahani-2010-ildn.pdf pg 10 [3] http://papers.nips.cc/paper/6477-understanding-probabilistic-sparse-gaussian-process-approximations [4] Titsias 2009

[Author Response · NeurIPS 2019]

**Reviewer #1**: We thank you for appreciating our contributions and providing valuable feedback, which will be taken into account when revising our paper, such as shortening Sect. 3 to free up space for including more details in Sect. 4.

The empirical results comparing parameter tying vs. naive design are in fact reported in Table 3 of Appendix C.2; a reader is referred to these results in lines 161 to 163 of the main paper. Based on your suggestion, we will move such a comparison from the appendix to the main paper.

Regarding how $\epsilon$ is shared among all the layers, observe from the left-hand side of Fig. 2d as well as from line 156 that the same $\epsilon$ is fed as an input to $g_{\phi_\ell}$ in each layer $\ell$ for $\ell = 1, \ldots, L$.

**Reviewer #4**: We thank you for providing insightful comments and advice, which will be incorporated into our revised paper. We will include a further discussion on how the other works of implicit VI (Titsias and Ruiz, 2019; Yin and Zhou, 2018) are related to IPVI, as you have suggested.

**Reviewer #5**: We thank you for providing valuable suggestions and feedback, which we will consider seriously in revising our paper. We would like to address your comments and questions below.

Regarding the necessity of parameter tying, we think overfitting is still an issue to be addressed. To see this, note that the generator's parameters (i.e., variational parameters) are trained to come close to the true posterior that, importantly, is conditioned on only the *training* data and not the entire data distribution (which encompasses the training data as well as the test data). Hence, variational inference (VI) is still subject to overfitting, even though it is regularized by the KL term. This motivates our use of parameter tying. We provide some experimental evidence below, as you have suggested.

Table I reports the train/test mean log-likelihood (MLL) achieved by IPVI with and without parameter tying for 2 small datasets: Boston ($N = 506$) and Energy ($N = 768$). For Boston dataset, it can be observed that no tying consistently yields higher train MLL and lower test MLL, hence indicating overfitting. This is also observed for Energy dataset when the no. of layers exceeds 2. For both datasets, as the no. of layers (hence no. of parameters) increases, overfitting becomes more severe for no tying. In contrast, parameter tying alleviates the overfitting issue considerably. There is no issue of overfitting in the discriminator, as you have correctly pointed out; we will clarify this in our revised paper.

Table I. Train/test mean log-likelihood (MLL) achieved by IPVI with and without parameter tying over 10 runs.

| Dataset | Boston ($N = 506$) | | | | |
|---|---|---|---|---|---|
| DGP Layers | 1 | 2 | 3 | 4 | 5 |
| No Tying | -1.86/-2.21 | -1.76/-2.37 | -1.64/-2.48 | -1.52/-2.51 | -1.51/-2.57 |
| Tying | -1.91/-2.09 | -1.79/-2.08 | -1.77/-2.13 | -1.84/-2.09 | -1.83/-2.10 |
| Dataset | Energy ($N = 768$) | | | | |
| DGP Layers | 1 | 2 | 3 | 4 | 5 |
| No Tying | -0.12/-0.44 | 0.03/-0.31 | 0.18/-0.34 | 0.20/-0.47 | 0.21/-0.58 |
| Tying | -0.16/-0.32 | -0.11/-0.34 | -0.02/-0.23 | 0.10/-0.01 | 0.17/0.13 |

About the performance gap of SGPs, the optimal variational posterior is indeed a Gaussian for single-layer SGP regression (Titsias, 2009). However, since the SGP model hyperparameters are not known beforehand, DSVI SGP has to *jointly* optimize its hyperparameters and variational parameters. Such an optimization is not convex. Hence, there is no guarantee that it will reach the global optimum. Thus, the performance gap can be explained by IPVI's ability to jointly find "better" values of hyperparameters and variational parameters. Additionally, in our experiments, the performance of DSVI has already converged within 20000 iterations and is thus not limited.

We have also computed the estimate of ELBO (by, after training, continuing to train the discriminator for longer), like you have suggested. Table II shows the mean ELBOs of DSVI and IPVI over 10 runs for the Boston dataset. IPVI generally achieves higher ELBOs, which agrees with results of the test log-likelihood in Fig. 4 of the main paper. Since SGHMC DGP is not based on VI, no ELBO is computed for that method; a comparison of its expressive power and performance with that of IPVI is already investigated and analyzed in the synthetic experiment in Section 5.1.

Based on your comments for the MNIST experiment, we increase the number of inducing points to 800. Table III reports the mean test accuracy over 5 runs. The results are consistent with that in Table 2 of the main paper where IPVI DGP 4 performs the best. Regarding the code, we will cite GPflow in our revised paper.

Table II. Mean ELBOs for Boston dataset.

| Model | DSVI | IPVI |
|---|---|---|
| SGP | -956.57 | -934.07 |
| DGP 2 | -850.54 | -846.65 |
| DGP 3 | -836.13 | -846.45 |
| DGP 4 | -787.10 | -776.93 |
| DGP 5 | -770.67 | -758.42 |

Table III. Mean test accuracy (%) on MNIST datasets.

| Dataset | MNIST | |
|---|---|---|
| | SGP | DGP 4 |
| DSVI | **97.92** | 98.05 |
| SGHMC | 97.07 | 97.91 |
| IPVI | 97.85 | **98.23** |

[Meta-Review · NeurIPS 2019]

As all reviewers agreed this is a solid paper presenting an interesting new inference method for Deep GPs. For the final version please address all reviewers' comments.